# ReGS: Reference-based Controllable Scene Stylization with Gaussian Splatting

**Yiqun Mei**[*]   **Jiacong Xu**[*]   **Vishal M. Patel**
Johns Hopkins University
{ymei7,jxu155, vpatel36}@jhu.edu

## Abstract

Referenced-based scene stylization that edits the appearance based on a content-aligned reference image is an emerging research area. Starting with a pretrained neural radiance field (NeRF), existing methods typically learn a novel appearance that matches the given style. Despite their effectiveness, they inherently suffer from time-consuming volume rendering, and thus are impractical for many real-time applications. In this work, we propose ReGS, which adapts 3D Gaussian Splatting (3DGS) for reference-based stylization to enable real-time stylized view synthesis. Editing the appearance of a pretrained 3DGS is challenging as it uses *discrete* Gaussians as 3D representation, which tightly bind appearance with geometry. Simply optimizing the appearance as prior methods do is often insufficient for modeling continuous textures in the given reference image. To address this challenge, we propose a novel texture-guided control mechanism that adaptively adjusts local responsible Gaussians to a new geometric arrangement, serving for desired texture details. The proposed process is guided by texture clues for effective appearance editing, and regularized by scene depth for preserving original geometric structure. With these novel designs, we show ReGS can produce state-of-the-art stylization results that respect the reference texture while embracing real-time rendering speed for free-view navigation.

## 1 Introduction

Stylizing a 3D scene based on a 2D artwork is an active research area in both computer vision and graphics [1, 2, 3, 4, 5, 6, 7]. One important direction of stylization aims to precisely stylize the scene appearance based on a 2D content-aligned reference image drawn by users [10]. Such problem has numerous applications in digital art, film production and virtual reality. In the classical graphics pipeline, completing this task requires experienced 3D artists to manually create a UV texture map as input to the shader, a tedious process requiring professional knowledge, significant time, and effort.

Over the past decades, tremendous progress has been made in automatic scene stylization by leveraging view synthesis methods. While early attempts [1, 2, 12, 13] suffer from geometry errors of point clouds or meshes, more recent methods [9, 8, 3, 4, 5, 6, 7] rely on radiance field (NeRF) [14], a powerful implicit 3D representation, to deliver high-quality renditions that are perceptually similar to the reference image. A typical stylization workflow starts from a pretrained NeRF model of the target scene, followed by an appearance optimization phase to match the given style. The density function is always fixed to maintain the scene geometry [8, 9, 3, 10, 6] . Despite their promising results, NeRF-based approaches consume high training and rendering costs in order to obtain satisfactory results. Although some recent efforts make fast training possible [15, 16, 17, 18, 19, 20], the improvement in efficiency often comes at the price of degraded visual quality. Meanwhile, real-time rendering at inference time still remains challenging.

---

[*]Equal contribution

38th Conference on Neural Information Processing Systems (NeurIPS 2024).

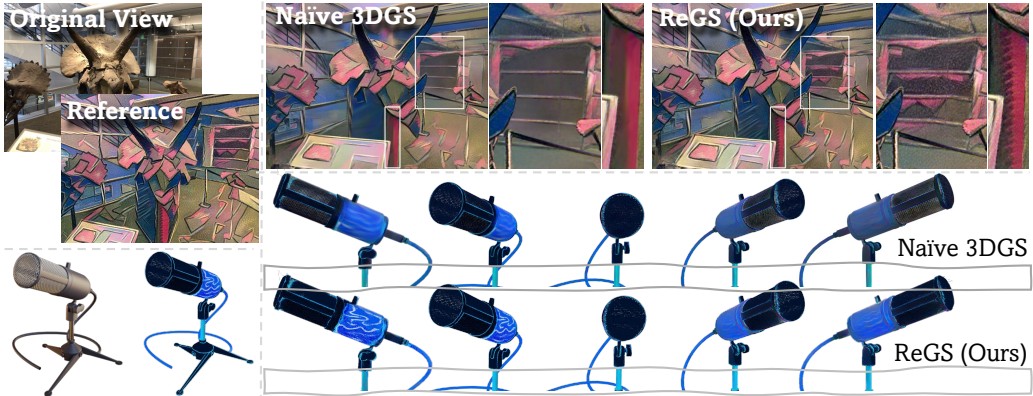

Figure 1: Given a pretrained 3DGS model of the target scene and its paired style reference, ReGS enables real-time stylized view synthesis (at 134 FPS) with high-fidelity texture well-aligned with the reference. In contrast, only optimizing the appearance of 3DGS (denoted as Naive 3DGS), as previous methods [8, 9, 3, 10, 6] do, fails to capture many texture details in the reference. We tackle the challenges in high-fidelity appearance editing with a texture-guided control mechanism that is significantly more effective than the default density control [11] in addressing texture underfitting. Side-by-side comparisons with default density control can be found in Figure 5.

Recently, 3D Gaussian Splatting (3DGS) [11] has become an emerging choice for representing 3D scenes. 3DGS creates millions of colored Gaussians with learnable attributes to jointly represent the target scene geometry and appearance. Importantly, it adopts splatting-based rasterization [21] to replace the time-consuming volume rendering of NeRF models, providing remarkably faster rendering speed while maintaining comparable visual quality. However, as it uses *discrete* 3D Gaussians to represent the reconstructed scene, optimizing their appearance with a fixed geometry layout (as NeRF-based methods do) is often inadequate to capture the *continuous* texture variance in the reference image. This "appearance-geometry entanglement" makes applying 3DGS to applications that require novel appearance, *i.e.* stylization, challenging. For 3DGS, how to properly control and edit the appearance without distorting the original geometry remains under-explored.

In this paper, we present a novel reference-based scene stylization method using 3DGS, dubbed ReGS, to enable real-time stylized view synthesis with high-fidelity textures well-aligned with the given reference. Similar to previous methods, our approach starts with a pretrained 3D Gaussian model of the target scene. The core enabler of ReGS is a novel texture-guided control procedure that makes high-fidelity appearance editing with ease. In particular, we adaptively adjust the local arrangement of responsible Gaussians in the appearance underfitting regions to a state that the desired textures specified in the reference image can be faithfully expressed. The control process is designed to **(1)** automatically identify target local Gaussians using texture clues, and **(2)** structurally distribute tiny Gaussians for fast detail infilling while **(3)** sticking to the original scene structure via a depth-based regularization. With these novel designs, ReGS is able to learn consistent 3D appearance that accurately follows the given reference image.

Following [10], we train ReGS on a set of pseudo-stylized images for view consistency, which are synthetic multi-view data created using extracted scene depth, alongside with a template-based matching loss to ensure style spread to the occluded regions. By combining these techniques with the proposed texture-guided control, ReGS is capable of producing visually appealing stylization results that attain both geometric and perceptual consistency. Through extensive experiments, we demonstrate that ReGS achieves state-of-the-art visual quality compared to existing stylization methods while enabling real-time view synthesis by embracing the fast rendering speed of Gaussian Splatting.

## 2 Related Work

### 2.1 3D Scene Representation

**Neural Radiance Field.** Reconstructing 3D scene from multi-view collections is a long-standing problem in computer vision. Early approaches adopting explicit mesh [22, 23, 24, 25] or voxel [26,

27, 28] based representations often suffer from geometry error and lack of appearance details [29]. Recent methods [30, 31, 32, 33, 34, 35] adopt learnable radiance fields [14] to capture 3D scene implicitly and outperform previous techniques by a large margin. However, NeRF models require millions of network queries for a single rendition that can be extremely time and resource-consuming. To reduce the training time, advanced methods adopt explicit/hybrid representations including voxel grid [18, 16, 15, 36, 37], octree [38, 39, 40], planes [41, 42, 17, 43] and hash grid [20], and successfully reduce the training time from days to minutes. Nevertheless, the rendering speed at inference time is still limited by their volumetric nature, which requires dense sampling along a ray to generate a single pixel.

**3D Gaussian Splatting**. Recently, 3D Gaussian Splatting (3DGS) [11] achieves real-time novel view synthesis based on a differentiable rasterizer [21] that efficiently projects millions of 3D Gaussians to a 2D canvas. Given its high efficiency, 3DGS becomes a promising solution to enable real-time vision applications, such as human avatar [44, 45, 46, 47], 3D object and immersive scene creation [48, 49, 50, 51], relighting [52, 53, 54, 55], surface or mesh reconstruction [56, 57], 3D segmentation [58, 59, 60], and SLAM [61, 62, 63]. Motivated by its high efficiency, our work explores 3DGS to enable real-time stylized view navigation.

## 2.2   2D Stylization

**Arbitrary Style Transfer.** Our method is related to the general 2D stylization [64], which transfers the style from an artwork to a target image while maintaining the original content structure. In the pioneering work, Gatys *et al*. [65] introduce an iterative scheme that progressively reduces the difference between the Gram statistics of generated image and style image features, yet lengthy optimization is required per style. To improve efficiency, later methods [66, 67, 68, 69, 70] focus on arbitrary image/video stylization by transferring the content image to target style spaces in a zero-shot manner. For example, Huang *et al*. [67] introduce AdaIN, which achieves real-time stylization by matching content features with the mean and standard deviation of style features. Linear style transfer [66] instead predicts a linear transformation matrix based on both content and style pairs. For video stylization, it is crucial to maintain temporal coherence of the stylized frames. Techniques [71, 72, 73, 74, 75, 76], such as flow-based wrapping [72], global SSIM constraint [75], and inter-frame feature similarity [76], are proposed to ensure the consistency.

**Optimization-based Style Transfer.** While arbitrary style transfer is desirable in terms of flexibility, they often fall short of reproducing small stylistic patterns and lack high-frequency details [8, 77]. Optimization-based Stylization [78, 79, 80, 81, 82, 77] is still the primary choice to ensure visual quality. For instance, a coarse-to-fine strategy is proposed by Liao *et al*. [82] to compute the nearest-neighbor field and build a semantically meaningful mapping between input and style images for visual attribute transfer. Kolkin *et al*. [77] reach state-of-the-art stylization quality by replacing the content features with the nearest style feature. To enable better controllability, example-based methods [83, 84, 85, 86] perform wrapping or stylizing based on the aligned correspondences between the style reference and content images. However, their 2D alignment is generally unsuitable for 3D scenes due to occlusions, leading to flickering effects [10].

## 2.3   3D Stylization

3D scene stylization extends artistic works beyond the 2D canvas [87]. To stylizing a 3D scene, both image exemplar [8, 9, 4] and text instructions [88, 89, 90, 91] have been explored as style guidance. This work focuses on image-exemplar-based methods. Early works [1, 2] typically back-project image colors as 3D point cloud for processing, and project stylized point features back to 2D for view synthesis. Yet, using point cloud often fails to represent complicated geometry and produces artifacts for complex scenes [8].

Benefiting from NeRF, methods stylizing radiance fields [9, 8, 3, 4, 5, 6, 7] have shown visually compelling and geometry-consistent results than previously possible. Similar to image stylization, several works [9, 3, 6, 7] deal with arbitrary or multiple style transfer using various techniques such as 2D-3D mutual learning [3], deferred style transformation [9], and hypernetwork [6]. While a universal stylizer might be desirable, these methods can only transfer the overall color tone and lack detailed style patterns, *i.e.* brushstrokes. Per-style optimization is still required for better visual quality. Among these methods [4, 8, 5, 92], ARF [8] shows state-of-the-art stylization capability

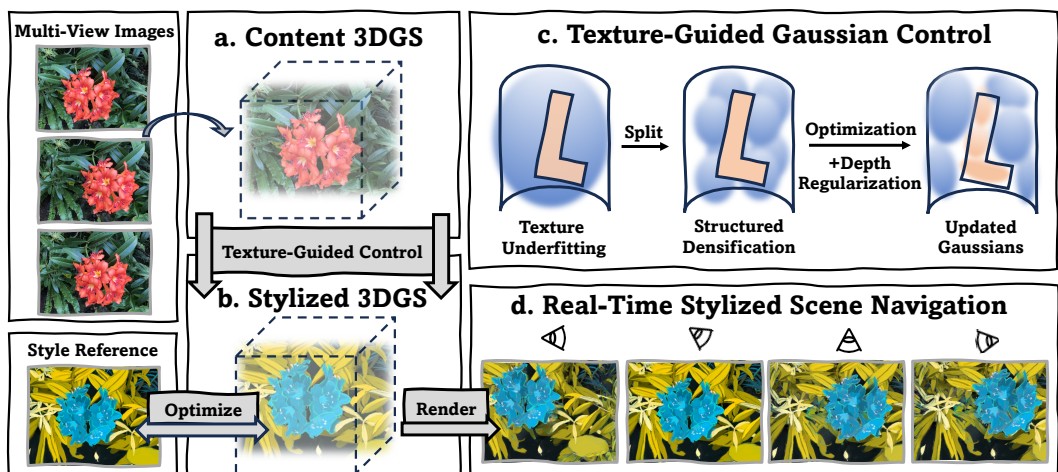

Figure 2: **An overview of ReGS**. (a) The proposed method starts with a pretrained content 3DGS of the target scene, and (b) outputs a stylized 3DGS that follows the reference. (c) We propose Texture-Guided Gaussian Control that can progressively resolve texture underfitting by automatically locating responsible Gaussians and adjusting local geometry layout for fitting high-frequency textures. (d) Once training is done, our method enables real-time stylized scene navigation.

by progressively matching the generated features with the closest style feature via nearest neighbor search. Besides NeRF-based approaches, concurrent works [93, 94, 95, 96] also explore 3DGS [11] as scene representation. However, these methods are designed for transferring styles from an arbitrary reference and lack controllability over generated results. To this end, Ref-NPR [10] introduces a reference-based scheme that controls stylized appearance based on a content-aligned reference image. Our work also focuses on this setting.

## 3 Method

An overview of ReGS is shown in Figure 2. ReGS takes a pretrained 3DGS model (Figure 2 (a)) of the target scene as well as a content-aligned reference image as inputs. It outputs a stylized 3DGS model (Figure 2 (b)) that bakes the texture of the reference image into the scene and enables real-time stylized views synthesis (Figure 2 (d)).

As 3DGS represents a scene as discrete Gaussians, simply optimizing its appearance often cannot capture the continuous texture details in the reference image. We introduce a texture-guided control mechanism to progressively address this challenge (Sec. 3.2). To ensure no geometry distortion happens during optimization, we propose a geometry regularization using scene depth (Sec. 3.3). We then introduce two techniques to encourage perceptual-consistent renditions (Sec. 3.4). Finally, we describe our training objectives in Sec. 3.5.

### 3.1 Preliminary: 3D Gaussian Splatting

Before introducing our method, we first provide a brief review of 3D Gaussian Splatting [11]. 3DGS represents the scene explicitly by a collection of learnable Gaussians. Each 3D Gaussian is attributed by a positional vector $\mu \in \mathbb{R}^3$ and a 3D covariance matrix $\Sigma \in \mathbb{R}^{3 \times 3}$. Its influence on a space point $\mathbf{x}$ is proportional to a Gaussian distribution:

$$G(\mathbf{x}) = e^{-\frac{1}{2}(\mathbf{x}-\mu)^\top \Sigma^{-1}(\mathbf{x}-\mu)}. \tag{1}$$

By definition, the covariance matrix should be positive semi-definite. This is achieved by decomposing $\Sigma$ into a scaling matrix $\mathbf{S}$ and a quaternion $\mathbf{R}$ *i.e.* $\Sigma = \mathbf{R}\mathbf{S}\mathbf{S}^\top\mathbf{R}^\top$. Each Gaussian also stores an opacity value $\alpha_i$ and a view-dependent color represented by Spherical Harmonic (SH) coefficients.

The rendering procedure is implemented as splatting-based rasterization [21] which projects Gaussians to a 2D canvas. The projected 2D splats are then sorted based on the depth to the camera. After

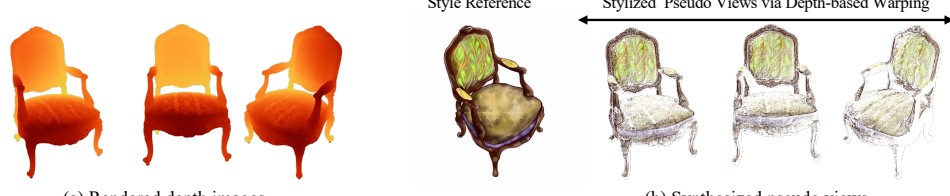

Style Reference          Stylized Pseudo Views via Depth-based Warping

(a) Rendered depth images                    (b) Synthesized pseudo views

Figure 3: Examples of (a) rendered depth maps using Eq.3 and (b) synthesized stylized pseudo views.

sorting, the final color for each pixel is computed through $\alpha$-blending:

$$C = \sum_{i=1}^{n} c_i \alpha'_i \prod_{j=1}^{i-1} (1 - \alpha'_j), \tag{2}$$

where $c_i$ is a view-dependent color of the $i$-th Gaussian computed from SH. $\alpha'_i$ is the multiplication result of the learned opacity $\alpha_i$ and evaluated value of the projected 2D Gaussian.

During optimization, heuristic controls are employed to adaptively manage the density of Gaussians to better represent the scene. Specifically, it densifies Gaussians with large positional gradients to capture missing geometry and prunes Gaussians with small opacity to improve compactness.

## 3.2    Texture-Guided Gaussian Control

As a discrete scene representation, the geometry layout and arrangement of Gaussians essentially limit the range of appearance it can express. For example, as shown in Figure 2 (c), appearance underfitting happens frequently at the area where local granularity of Gaussians is greater than the variance of the texture, *e.g.* a smooth colored surface in the original scene is painted with rich details in the reference view. ReGS addresses such challenges via a novel texture-guided control that splits these responsible local Gaussians into a denser set suitable for high-frequency texture. Specifically, the proposed mechanism automatically identifies responsible Gaussians using texture clues and structurally replaces them with a denser set of tiny Gaussians to compensate for the missing details. We describe important designs of the proposed algorithm below.

**Texture Guidance.**  The control algorithm is directly guided by texture clues. Specifically, we accumulate color gradients of all Gaussians over iterations and select Gaussians with larger gradient magnitude than a threshold for densification. We found that a larger color gradient corresponds to Gaussians that have large texture errors while the optimization struggles to find the correct colors to reduce the loss. This control scheme shares a similar spirit with the original control scheme in 3DGS, where they leverage positional gradients to locate Gaussians responsible for missing geometric features. But in stylization, scene geometry is already well-reconstructed through pretraining, and therefore, the positional gradient is no longer informative. As demonstrated in Figure 5, our color-based control scheme is more sensitive for pinpointing Gaussians with missing textures than the positional-based solution. In practical implementation, we increase density based on the gradient statistics of every 100 iterations.

**Structured Densification.** Traditional mesh subdivision [97] cuts large faces into more sub-faces to express surface details. Sharing a similar spirit, we structurally split each responsible Gaussians into a structured denser set to better represent texture details. Intuitively, after densification, newly added Gaussians need to approximate the original space coverage to avoid inducing large geometry errors, and they should be sufficiently small and form a dense set to capture appearance variance. Based on these considerations, we propose a structured densification scheme that adds tiny Gaussians into the most representative locations surrounding their parent Gaussian. Specifically, we use nine tiny Gaussians to replace a parent Gaussian. Eight of them correspond to eight separate octants divided by the equatorial plane and perpendicular meridian planes of the original ellipsoid. And the rest is placed at the original center. We reduce their size by shrinking the scales with a factor of 8 and copy remaining parameters from their parent Gaussian. We empirically found this setup can roughly maintain a space coverage that approximates the original geometry. As optimization continues, the densified Gaussians are progressively updated to infill missing textures.

### 3.3 Depth-based Geometry Regularization

While our control mechanism progressively improves texture details, it is essential to ensure the original scene geometry is preserved after optimization. We resort to the scene depth as an additional regularization to penalize geometry changes. Examples of rendered depth are shown in Figure 3 (a). Formally, we derive the scene depth via a $\alpha$-blending-based equation:

$$d = \sum_{i=1}^{n} d_i \alpha_i' \prod_{j=1}^{i-1} (1 - \alpha_j'), \tag{3}$$

where the $d_i$ is the z-buffer associated with the $i$th Gaussian and $\alpha_i'$ is the same evaluated opacity in Eq. 2. $d_i$ is computed by projecting the 3D location $\mu$ to the camera space.

The depth regularization is defined as the $L_1$ distance between a depth image $D_i$ rendered from original scene model $\mathbf{m}$ and a depth image $\widehat{D}_i$ rendered from the stylized model $\widehat{\mathbf{m}}$ using the same camera pose $\phi_i$ $i.e.$ $\mathcal{L}_{depth} = \|\widehat{D}_i - D_i\|_1$.

### 3.4 View-Consistent Stylization

For stylization, it is necessary to ensure the stylized appearance is consistent across different viewpoints and inpaints the occluded areas. Following [10], we adopt two strategies to address them.

**Stylized Pseudo View Supervision.** An image with paired depth contains sufficient information to re-render from nearby viewpoints [98]. This allows us to create a set of stylized pseudo views for obtaining additional supervision from the reference image. Our synthesis approach is very similar to classic depth-based 3D warping [98, 99]. Specifically, we back-project the reference image $S_R$ to the world space using the depth image $D_R$ and its camera pose $\phi_R$. Then, we re-project these 3D points back to a new viewpoint $\phi_i$. The resulting 2D image $S_i$ is used as an additional style supervision. Examples of the created pseudo views are shown in Figure 3 (b). It is important to make sure supervision only happens on meaningful pixels, $i.e.$ they are projections of 3D points that are visible from the current viewpoint $\phi_i$. Therefore, we conduct a visibility check by comparing the depth between the 2D projections of the 3D points and the depth image $D_i$ from the current viewpoint $\phi_i$. This results in a visibility mask $M_i$. Given the pseudo views and visibility masks, one can define a pseudo view supervision loss as

$$\mathcal{L}_{view} = \frac{1}{\|M_i\|_0} \|M_i \widehat{S}_i - M_i S_i\|_1, \tag{4}$$

where $\|.\|_0$ is the $\ell_0$-norm that counts the number of valid pixels and $\widehat{S}_i$ is renderings of the stylized model $\widehat{\mathbf{m}}$.

**Template Correspondence Matching (TCM) Loss.** To ensure stylized appearance spreads to the occluded areas, we adopt the same TCM loss proposed in [10]. We briefly describe it here and refer readers to [10] for more details. TCM regularizes the difference of semantic correspondences before and after stylization. Given the style reference $S_R$, its corresponding view $I_R$, and a scene image $I_i$ rendered from a camera pose $\phi_i$, it constructs a guidance feature $F_G$ by $F_{G_i}^{(x,y)} = F_{S_R}^{(x^*,y^*)}$ where

$$(x^*, y^*) = \underset{x',y'}{\operatorname{argmin}} \, \mathbf{dist}(F_{I_i}^{(x,y)}, F_{I_R}^{(x',y')}). \tag{5}$$

Here, $F_{S_R}, F_{I_R}, F_{I_i}$ denote deep semantic features of image $S_R$, $I_R$, and $I_i$ extracted by an ImageNet pretrained VGG [100]. The superscript $(x, y)$ denotes the $xy$ coordinates on the feature map. **dist** denotes the cosine distance. After obtaining the guidance feature, TCM loss is defined as a cosine distance loss:

$$\mathcal{L}_{TCM} = \mathbf{dist}(F_{\widehat{s}_i}, F_{G_i}), \tag{6}$$

where $F_{\widehat{s}_i}$ is the extracted VGG features of the generated stylized view $\widehat{S}_i$.

### 3.5 Training Objectives

Besides aforementioned depth loss $\mathcal{L}_{depth}$, pseudo view supervision loss $\mathcal{L}_{view}$ and TCM loss $\mathcal{L}_{TCM}$, ReGS further optimizes a reconstruction loss $\mathcal{L}_{rec}$ and a coarse color-matching loss $\mathcal{L}_{color}$ [10]. The reconstruction loss is defined as the $L_1$ distance between the reference $S_R$ and the corresponding

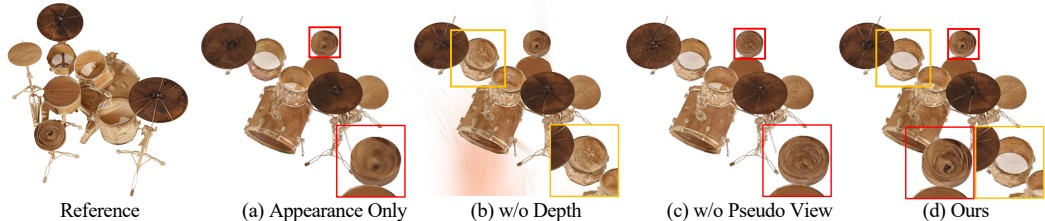

| Reference | (a) Appearance Only | (b) w/o Depth | (c) w/o Pseudo View | (d) Ours |

Figure 4: **Ablation study on different components of ReGS**. (a) Optimizing only the appearance of a 3DGS model cannot reproduce texture details. (b) Removing depth regularization causes Gaussians to float out from the surface and distort the origin geometry. (c) Without pseudo-view supervision, results lack view consistency. (d) Our full model produces the best results that faithfully respect the texture in the reference.

stylized output $\hat{S}_R$ to enforce appearance baking. The color-matching loss is defined as

$$\mathcal{L}_{color} = \|\widehat{S}_i^{(x,y)} - S_R^{(x^*,y^*)}\|_2^2, \tag{7}$$

where $S^{(x,y)}$ denotes the average color of a patch associated with feature-level index $(x,y)$. Feature-level index is computed using Eq. 9. This loss is directly adapted from [10] to encourage overall color matching in the occluded area. The overall loss for ReGS can be expressed as

$$\mathcal{L} = \lambda_{rec}\mathcal{L}_{rec} + \lambda_{depth}\mathcal{L}_{depth} + \lambda_{view}\mathcal{L}_{view} + \lambda_{tcm}\mathcal{L}_{TCM} + \lambda_{color}\mathcal{L}_{color} \tag{8}$$

where $\lambda_{(.)}$ denotes the balancing parameter.

### 3.6 Implementation and Training Details

ReGS uses 3D Gaussians [11] as the scene representation and is built upon their official codebase. We follow the default parameter settings to obtain the pretrained 3D Gaussian model of the photo-realistic scene. For stylization, as we do not expect view-dependent effects, we discard the higher order SH and only render diffuse color in the stylization phase. Therefore, content images used in $\mathcal{L}_{TCM}$ and $\mathcal{L}_{color}$ are the results of this diffuse model.

For texture-guided control, we start accumulating gradients after a warm-up of 100 iterations and then perform the densification operation based on the color gradient statistics of every 100 iterations. The control process stops when it reaches half of the total iterations. The gradient threshold is empirically set to $1e-5$ at the beginning, and we linearly reduce it to $5e-6$ to allow for refining tiny details in the later training stage. Following [10, 8], we use the ImageNet pretrained VGG16 [100] as the feature extractor and use the features produced by *relu_3* and *relu_4* in $\mathcal{L}_{TCM}$. For balancing parameters we set $\lambda_{rec} = \lambda_{tcm} = 1$, $\lambda_{depth} = 10$, $\lambda_{view} = 2$, and $\lambda_{color} = 15$, which are determined by a simple grid search on Blender [14] scenes. At each iteration, we always sample two views: the reference view and a random view. We train our model for 3000 iterations. The proposed method is implemented using PyTorch and trained on one A5000 GPU.

## 4 Experiments

In this section, we demonstrate the stylization quality and our designs through extensive experiments. More experiment results and ablations can be found in the supplemental file and accompanied video.

### 4.1 Datasets

The only available reference-based stylization dataset is provided by [10]. The dataset contains 12 selected scenes from Blender [14], LLFF [101], and Tanks and Temples [102]. Each scene is paired with a content-aligned reference image.

### 4.2 Ablation Study

We conduct controlled experiments to analyze the effectiveness of each design choice in ReGS. Results are illustrated in Figures 4 & 5. As illustrated in Figure 4, replacing any components of ReGS will harm the stylization quality. For example, Figure 4 (a) shows that optimizing only the

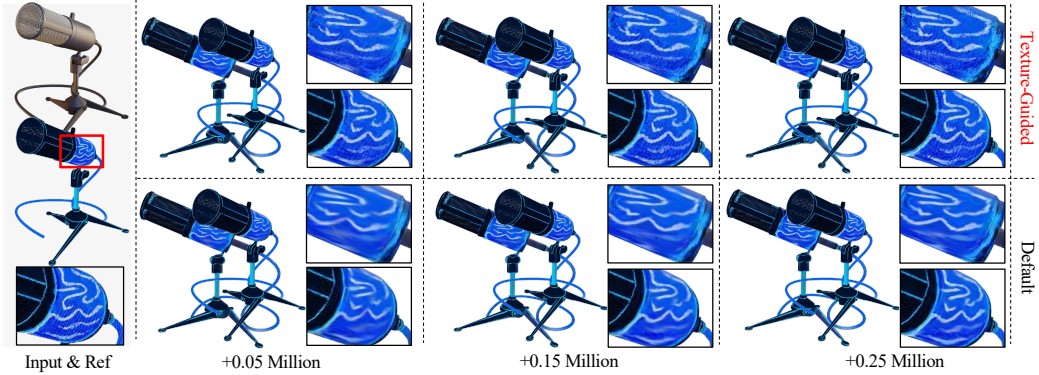

Figure 5: **Effectiveness of Texture-Guided Control.** We conduct controlled experiments by limiting the number of newly densified Gaussians throughout optimization. The pretrained model contains 0.3M Gaussians. The proposed texture-guided control can more faithfully reproduce the target texture details with a small number of Gaussians added (0.05M). The default strategy struggles to capture high-frequency details, even with a large number of Gaussians added (0.25M).

appearance with fixed geometry arrangement like previous methods [8, 6, 3, 9, 10] do fails to recover the texture details. As shown in Figure 4 (b), after removing depth regularization, Gaussians float out from the surface and distort the original scene geometry. Similarly, discarding the pseudo view supervision will induce view-inconsistency as highlighted in the inset (Figure 4 (c)). The full model overcomes these issues and produces more visually appealing results that follow the given reference.

**Effectiveness of Texture-Guided Control.** The core enabler of ReGS is the proposed texture-guided control mechanism that makes high-fidelity appearance editing with ease. Here, we demonstrate its effectiveness by comparing it with the default positional-gradient-guided density control [11] in addressing texture underfitting. Specifically, we conduct controlled experiments by setting a series of limits on the total number of Gaussians that can grow throughout optimization. Results are reported in Figure 5. One can see that by growing a very small amount of Gaussians (0.05M), the proposed texture-guided method can quickly infill most of the missing details. With more Gaussians added, it can further faithfully reproduce the given texture. In contrast, even with a large amount of new Gaussians (0.25M) created, the default method can barely capture high-frequency texture details. This is mainly because positional gradient is not sensitive to texture errors. As such, it fails to grow Gaussians in the regions with texture underfitting. And further moving these incorrectly placed Gaussians to the correct place for texture infilling is challenging. These results demonstrate our method is indeed more favorable for addressing appearance underfitting. Study on individual component (*i.e.* structured densification and color-gradient guidance) can be found in the supplement.

Table 1: Quantitative comparison of different stylization methods.

| Metric | ARF [8] | SNeRF [4] | Ref-NPR [10] | ReGS (Ours) |
|---|---|---|---|---|
| Ref-LPIPS↓ | 0.394 | 0.405 | 0.339 | **0.202** |
| Robustness↑ | 26.34 | 26.03 | 28.11 | **31.27** |
| Speed (fps) | 16.5 | 16.3 | 16.4 | **91.4** |

## 4.3 Compare with State-of-the-art Methods

To evaluate stylization performance, we compare our method with three state-of-the-art baselines: ARF [8], SNeRF [4], and Ref-NPR [10]. ARF [8] and SNeRF [4] are general stylization methods that conduct style transfer without considering content correspondence. Ref-NPR [10] is a reference-based stylization method similar to ours that aims to precisely edit the 3D appearance based on the reference. All baselines are NeRF-based approaches built upon Plenoxels [15].

**Qualitative Evaluation.** We report qualitative results in Figure 6. As shown, ARF [8] and SNeRF [4] cannot generate semantic-consistent results with respect to the reference image as they ignore content correspondence. In contrast, Ref-NPR [10] produces more controllable results but yields artifacts. In some challenging cases (*e.g.* last row in Figure 6), it also fails to achieve semantic consistent

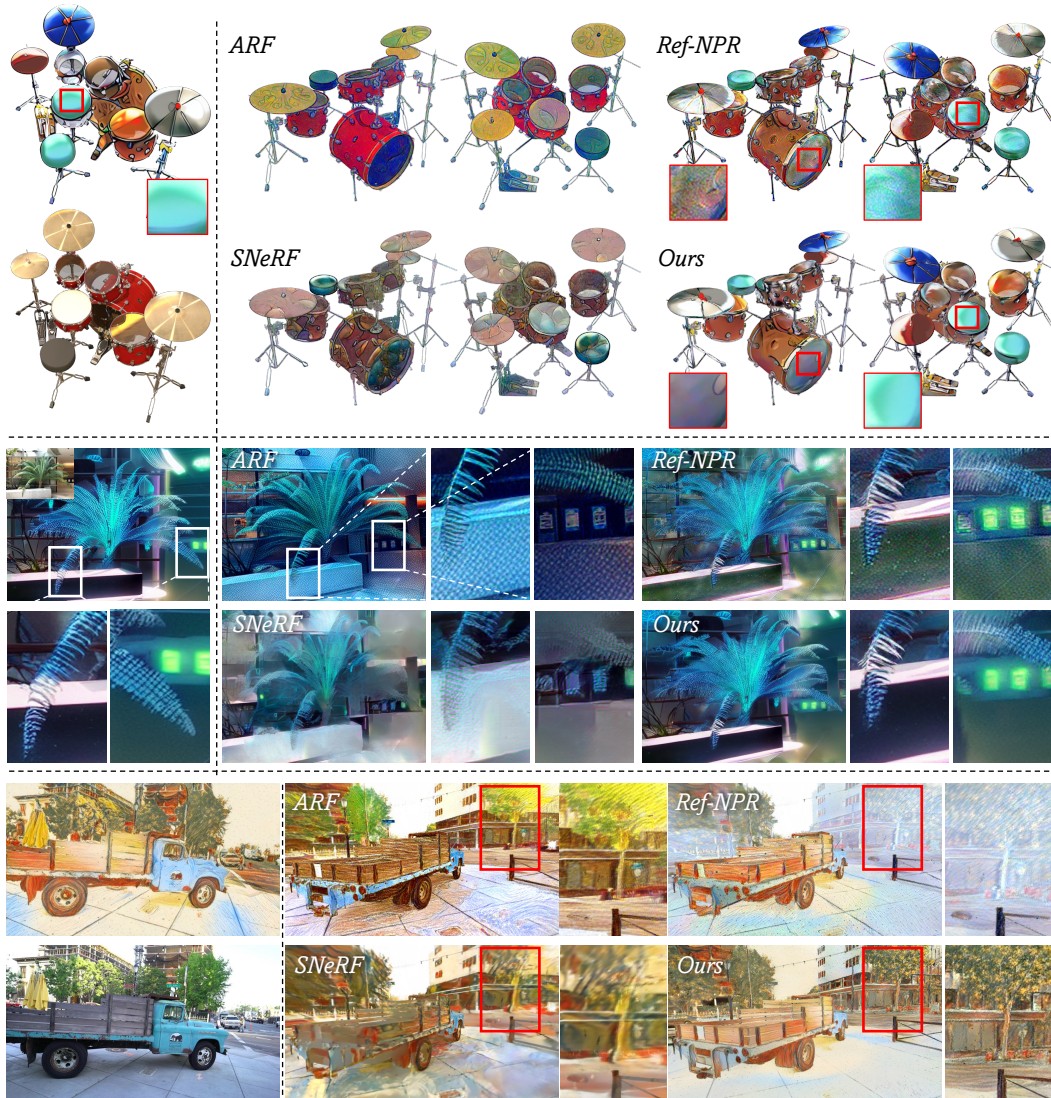

Figure 6: **Visual comparisons with state-of-the-art methods.** Paired reference and content view are shown on the left. Our method produces visual-compelling results that precisely follow the texture of the given reference, including the challenging high-frequency details such as the leaf in the second example. Baseline methods [10, 8, 4] either lack semantic consistency or produce artifacts.

stylization (*i.e.* green tree in the reference image is colored as white). In contrast, our method achieves better results that reproduce the desired texture, including challenging high-frequency ones.

**Quantitative Evaluation.** We present quantitative results in Table 1. Results are averaged over all scenes. We follow the protocol from [10] and report Ref-LPIPS and Robustness. Ref-LPIPS computes LPIPS [103] score between the reference image and the 10 nearest test views. To calculate robustness, we first (1) train a stylized base model $m_b$ and use it to render a set of stylized views as new references; (2) then we use these references to train another set of stylized models and (3) compute PSNR results between images produced by them and $m_b$ (using the same camera path). To measure run-time efficiency, we also report run-time FPS on a single A5000 GPU. As shown in Table 1, our method achieves the best results in terms of both quality and efficiency. Notably, our method enables real-time stylized view synthesis at 91 FPS.

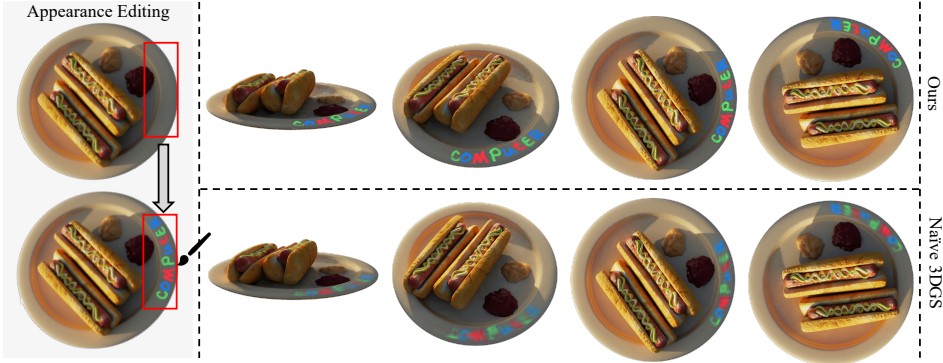

Figure 7: **Application: appearance editing.** Given a pretrained 3DGS model, our method allows users to make 3D appearance editing with ease by drawing on a 2D rendered view. Unlike NeRFs [14], just optimizing the appearance of a 3DGS model cannot robustly handle user edits.

## 5 Application: Appearance Editing

ReGS naturally enables high-fidelity appearance editing. As shown in Figure 7, given a pretrained 3DGS model and its rendering, our method allows users to make freehand edits on the image (*e.g.* "computer" on the plate) and robustly bake the edits back into the 3D scene with view-consistency. Unlike NeRFs [14], such task cannot be robustly handled by just optimizing the appearance of 3DGS (denoted as Naive Gaussian), especially when edits happen on a smooth surface where the granularity of Gaussians is greater than the texture variance. Benefiting from texture-guided control, our method can effectively locate these large Gaussians and replace them with a denser set for better appearance editing.

## 6 Conclusion

In this work, we introduce ReGS, which adapts Gaussian Splatting for reference-based controllable scene stylization. ReGS adopts a novel texture-guided control mechanism to make high-fidelity appearance editing with ease. This is achieved by adaptively replacing responsible Gaussians with a denser set to express the desired appearance details. The control process is guided by texture clues for appearance editing while preserving original scene geometry through a depth-based regularization. We demonstrate the state-of-the-art scene stylization quality and effective designs of ReGS through extensive experiments. Benefiting from the high efficiency of 3DGS, our method naturally enables real-time stylized view synthesis. Discussions of limitations can be found in the supplemental file.

## 7 Acknowledgment

This research is based upon work supported by the Office of the Director of National Intelligence (ODNI), Intelligence Advanced Research Projects Activity (IARPA), via IARPA R&D Contract No. 140D0423C0076. The views and conclusions contained herein are those of the authors and should not be interpreted as necessarily representing the official policies or endorsements, either expressed or implied, of the ODNI, IARPA, or the U.S. Government. The U.S. Government is authorized to reproduce and distribute reprints for Governmental purposes notwithstanding any copyright annotation thereon.

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

# Supplemental Material

## A  Video Demonstration

We encourage readers to watch the provided supplemental video for a better demonstration of the stylization quality of ReGS.

## B  More Implementation Details

We describe additional implementation details here. As discussed in the main manuscript, our method adopts the TCM loss $\mathcal{L}_{TCM}$ and color-matching loss $\mathcal{L}_{color}$ to encourage stylization spread to occluded areas. Specifically, similar to [10], we use the standard TCM loss for the first 70% iterations. For the last 30% iterations, we replace the content feature matching (Eq. 5 in the main manuscript) with a style feature matching, *i.e.* matching between features of a generated stylized view $\widehat{S}_i$ and the reference image $S_R$. This results in an online stylization loss similar to NNFM [8]. Formally, we construct an online guidance feature $F_{G_i}$ by $F_{G_i}^{(x,y)} = F_{S_R}^{(x^*,y^*)}$ where

$$(x^*, y^*) = \operatorname*{argmin}_{x',y'} \mathbf{dist}(F_{\widehat{S}_i}^{(x,y)}, F_{S_R}^{(x',y')}). \tag{9}$$

where $F_{\widehat{S}_i}$ and $F_{S_R}$ denote the deep semantic features of a generated stylized view $\widehat{S}_i$ and the reference image $S_R$ extracted by an ImageNet pretrained VGG [100]. And the loss is still computed as the cosine distance between $F_{\widehat{S}_i}$ and $F_{G_i}$. We further remove the color matching loss at this phase. These techniques are shown to be useful for a smoother content update [10]. For appearance editing, training takes about 50 seconds. For spreading appearance to other views, training takes about 5-6 minutes for all scenes due to the costly TCM loss. Detailed algorithms can be found in Algorithm 12.

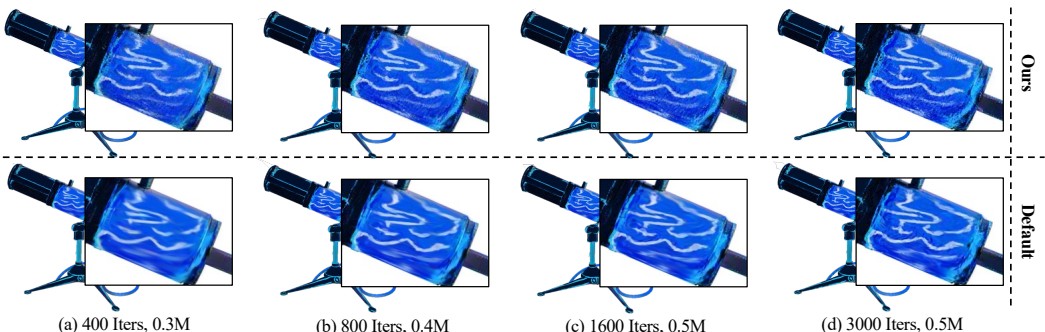

(a) 400 Iters, 0.3M    (b) 800 Iters, 0.4M    (c) 1600 Iters, 0.5M    (d) 3000 Iters, 0.5M

Figure 8: **Additional ablation study on structured densification.** Structured densification allows ReGS to create dense set of tiny Gaussians for representing high-frequency details. This enables our method to quickly infill the most of textures with a small amount of Gaussians tiny created ((a)). In contrast, the default strategy fails to express many details even with a large amount of Gaussians added ((d)).

## C  Additional Ablation Study

### C.1  Structured Densification

Structured densification replaces a responsible Gaussian by a dense set of tiny Gaussians without inducing large geometry errors. Here we investigate its effectiveness by comparing it with the default replacement strategy in [11] that splits a parent Gaussian into two "medium-sized" Gaussians by shrinking with a scale factor of 1.6. Compared to their method, our approach creates a much denser set (9 vs. 2 added Gaussians) of much smaller (shrinking by 8 vs. 1.6) Gaussians in a structured way to replace the parent Gaussian. We report the comparison results in Figure 8. Each column corresponds to a snapshot at the noted iterations during training. Densification stops at the 1500 iteration. For fair comparison, we reduce the gradient threshold of the default strategy to ensure that densification creates a similar number of new Gaussians after each operation. Both methods are

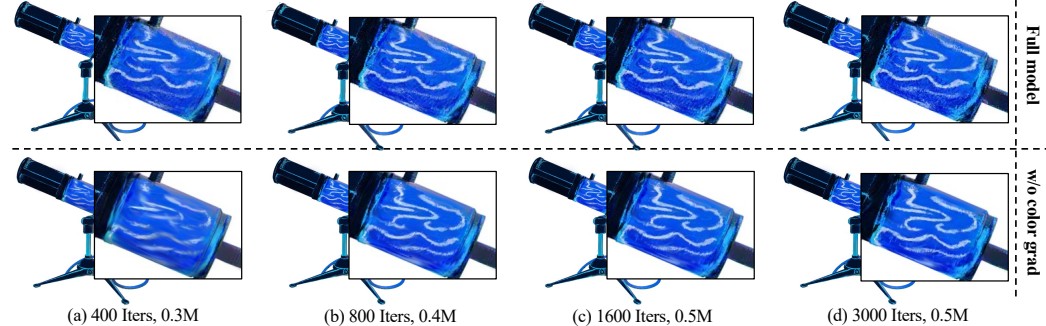

(a) 400 Iters, 0.3M    (b) 800 Iters, 0.4M    (c) 1600 Iters, 0.5M    (d) 3000 Iters, 0.5M

Figure 9: Ablation study on Texture Guidance (*i.e.* color-gradient guidance). Replacing texture guidance with positional-gradient guidance (bottom) fails to capture texture details.

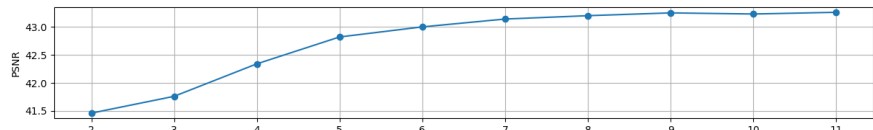

Figure 10: Ablation study on the number of Gaussians for each responsible Gaussian to be split. Small number cannot capture full details. Performance becomes saturated as the number grows.

based on the color-gradient-guided density control. As shown in Figure 8, the default strategy fails to express many details even after creating a large amount of Gaussians (*i.e.* Figure 8 (d)). This is mainly because the granularity of added Gaussians are not small enough to capture the high-frequency details. Therefore, their model has to repeatedly densify many times to reach a granularity that can match the texture variance. In contrast, benefiting from the structured densification, our method can quickly reproduce most of details by creating a small amount of tiny Gaussians, and shows much faster convergence. These results demonstrate the effectiveness of the proposed strategy in addressing texture underfitting.

## C.2    Texture (Color-Gradient) Guidance

Similarly, here we conduct an ablation study on color-gradient guidance. We construct the baseline by removing texture guidance from the full model (i.e., switching to the default positional-gradient guidance). We report the results in Figure 9. As shown, without texture guidance, the model fails to capture tiny texture details in the reference.

## C.3    Densification Number

Structured desertification splits a large Gaussian into a set of small Gaussians to infill the missing texture. Here we conduct an ablation study on the number of Gaussians for each responsible Gaussian to be split and present results in Figure 10. We plot the PSNR value between the style reference and the corresponding stylized view to quantitatively show the texture fitting capability using Blender scenes. As shown, when the number is small, the model fails to capture the target texture details. As this number grows, the performance becomes saturated. When the number equals 9, the model can

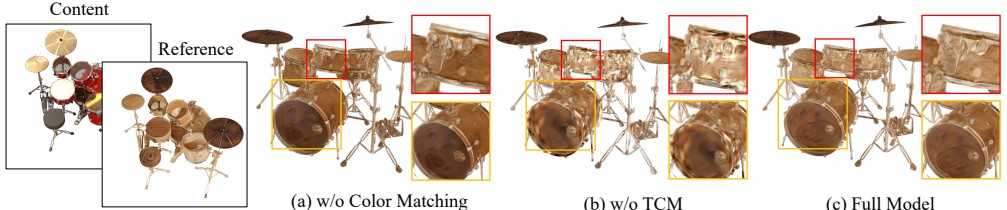

(a) w/o Color Matching    (b) w/o TCM    (c) Full Model

Figure 11: Ablation study on adopted TCM and color matching loss. Removing color matching loss leads to color mismatch and removing TCM leads to artifacts in occluded areas.

| **Algorithm 1:** Texture-Guided Control | **Algorithm 2:** Structured Densification (StructDensify) |
|---|---|
| **Input:** $\mathcal{G} = \{G_i(\mu_i, \alpha_i, c_i, s_i, r_i)\}_{i=1}^K$ current set of 3D Gaussians; $\nabla_c^i \mathcal{L}$ accumulated color gradient for $G_i$; $T_c$ densification threshold | **Input:** $G(\mu, \alpha, c, s, r)$ target Gaussian to be densified |

**Algorithm 1: Texture-Guided Control**

**Input:** $\mathcal{G} = \{G_i(\mu_i, \alpha_i, c_i, s_i, r_i)\}_{i=1}^K$ current set of 3D Gaussians; $\nabla_c^i \mathcal{L}$ accumulated color gradient for $G_i$; $T_c$ densification threshold

1 **for** $i \leftarrow 1$ **to** $K$ **do** *run in parallel*
2    **if** $\nabla_c^i \mathcal{L} > T_c$ **then**
3      run structured densification on $G_i$ as $\tilde{\mathcal{G}} = \text{StructDensify}(G_i)$;
4      update $\mathcal{G}$ as $\mathcal{G} = \mathcal{G} \cup \tilde{\mathcal{G}} \setminus \{G_i\}$;
5 **end**
6 **return** $\mathcal{G}$

**Algorithm 2: Structured Densification (StructDensify)**

**Input:** $G(\mu, \alpha, c, s, r)$ target Gaussian to be densified

1 **for** $i \leftarrow 1$ **to** $K = 9$ **do** *run in parallel*
2    compute placement location $\mu_i$ of the $i$-th new Gaussian; // `line 181`
3    initialize $\alpha_i, c_i, s_i, r_i$ from $\alpha, c, s/8, r$;
4    initialize $G_i$ as $G_i(\alpha_i, c_i, s_i, r_i)$;
5 **end**
6 **return** $\{G_i\}_{i=1}^9$

Figure 12: Algorithms of the proposed ReGS.

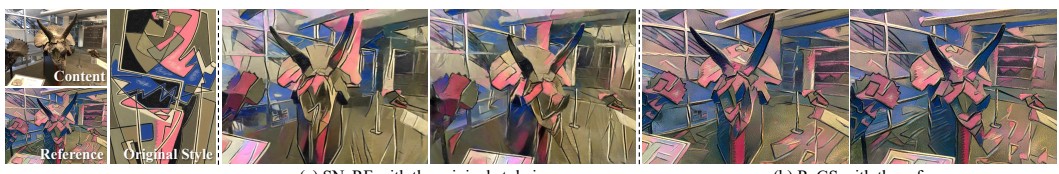

(a) SNeRF with the original style image      (b) ReGS with the reference

Figure 13: Visual comparison on aesthetic quality with SNeRF (using original style image).

achieve peak performance but also without inducing many excessive Gaussians that might slow down rendering.

### C.4 Additional Loss Components

ReGS adopts TCM and color matching loss from [10] to spread textures to occluded areas. Here we re-assess their effectiveness for 3DGS. As shown in Figure 11, using color matching loss reduces color mismatch and using TCM loss removes artifacts in the occluded areas. These findings are similar to the observations in [10]. These results suggest that the adopted losses indeed work for our 3DGS-based model.

## D    Limitations and Future Work

With ReGS, one can achieve real-time stylized view synthesis at high quality. However, it cannot significantly improve the training efficiency over previous methods [10, 8, 4], when adapting to a novel style reference. During training, efficiency bottleneck comes from the feature extraction and matching steps in $\mathcal{L}_{TCM}$, which is significantly slower than the splatting rendering [11]. Therefore, using 3DGS as scene representation cannot benefit much for the training efficiency. On the other hand, arbitrary stylization methods [9, 6] have made possible for adapting to a novel style in a zero-short manner with reasonable good performance. Following their spirit, designing a universal 3D Gaussian stylizer that can generalize to any references without run-time optimization might be an interesting direction for further improving training efficiency.

Moreover, beside editing the appearance, further styling geometry can be another interesting future direction. ReGS might be able to handle minor shape changes, for example, by relaxing the depth supervision. However, precise geometry editing based on a reference image is inherently more challenging due to single-view shape ambiguity. To achieve high-quality geometry stylization, existing methods often adopt a very different set of techniques such as shape prior [104], text guidance [105] and/or generative modeling [88, 90, 89] to hallucinate missing geometry. Combining our method with these techniques for joint geometry and appearance editing is an open and interesting future direction.

## E    Additional Comparison on Aesthetic Quality

In Figure 13, we provide additional comparisons on aesthetic quality with SNeRF [4], by providing the original 2D art image. One can see that SNeRF produces results mimicking the abstract style of

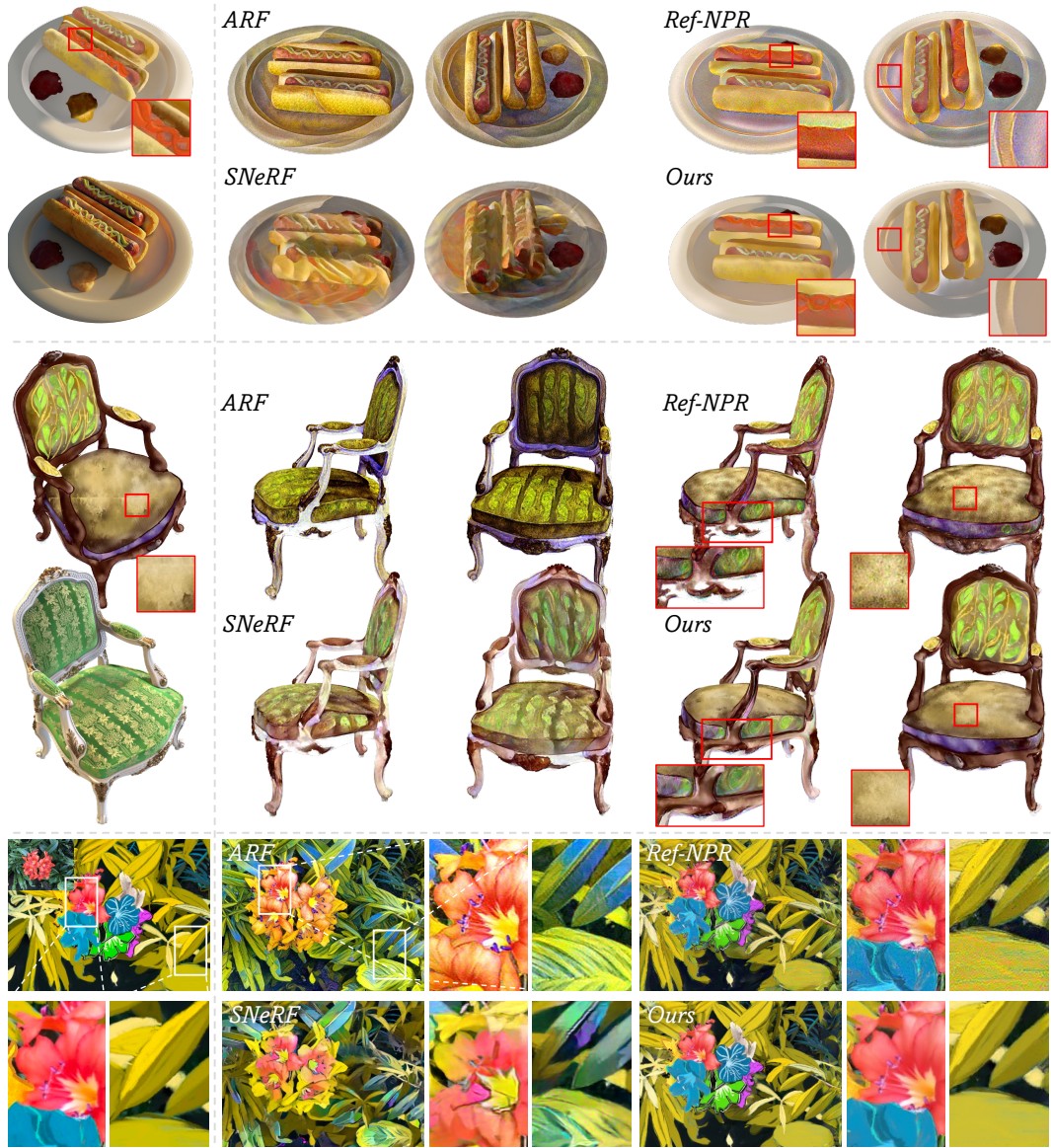

Figure 14: More visual comparison results. Our method can faithfully reproduce the texture in reference image.

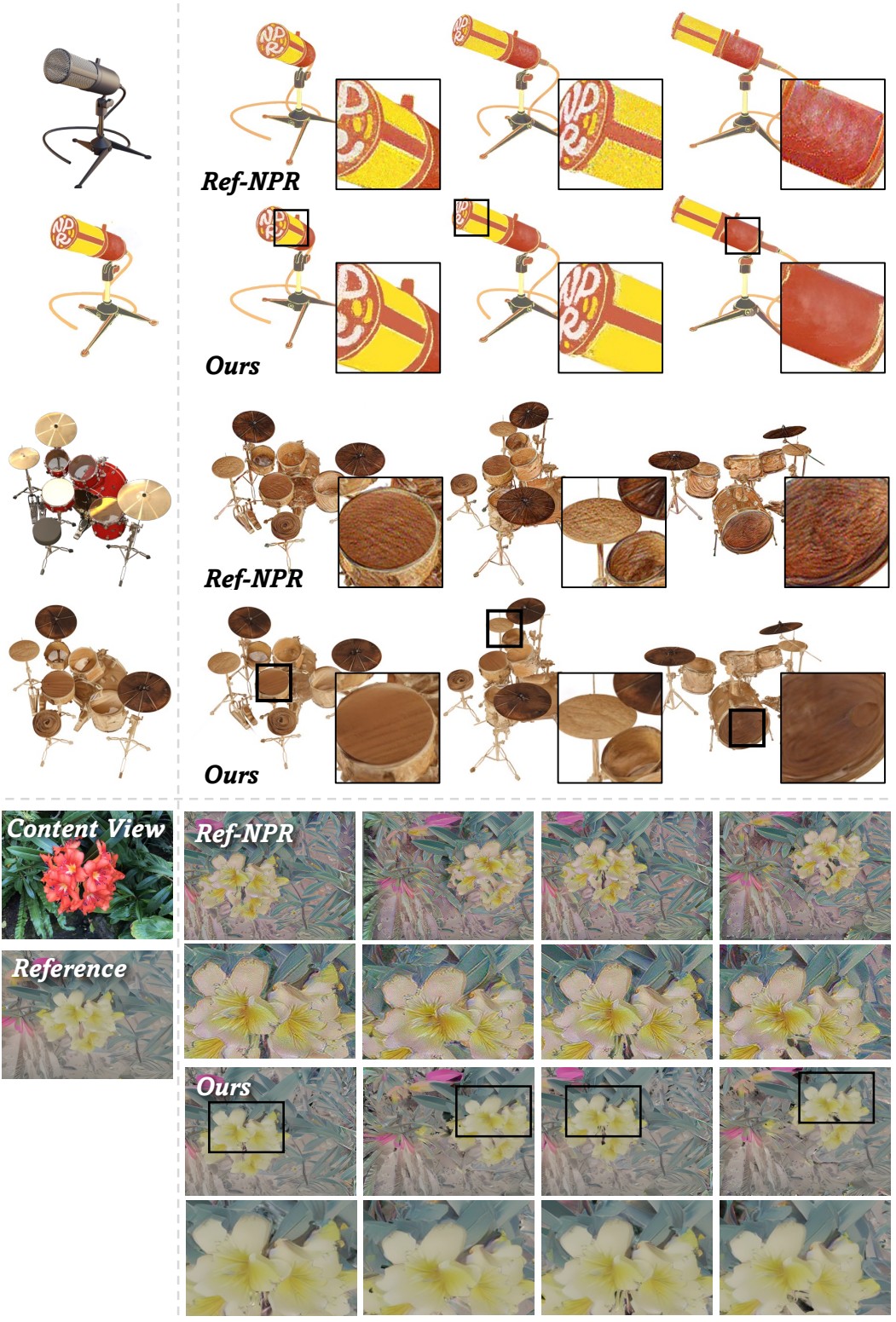

Figure 15: More visual comparison results with Ref-NPR [10]. Our method faithfully reproduce the texture in reference image. In contrast, Ref-NPR [10] produces images with lower quality.

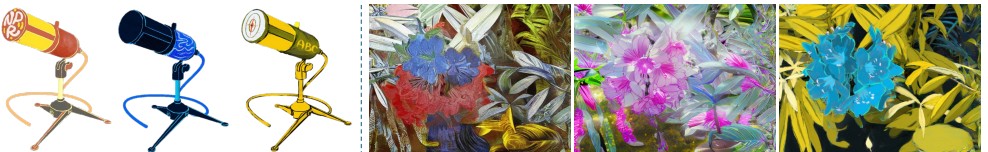

Figure 16: Stylization results of multiple style references on a single baked scene.

Table 2: Detailed Ref-LPIPS results on each scene

| Method | Chair | Ficus | Hotdog | Mic | Flower | Horn | Truck | Playground | Average |
|--------|-------|-------|--------|-----|--------|------|-------|------------|---------|
| ARF | 0.185 | 0.123 | 0.300 | 0.146 | 0.619 | 0.502 | 0.683 | 0.592 | 0.394 |
| SNeRF | 0.188 | 0.129 | 0.283 | 0.138 | 0.646 | 0.492 | 0.702 | 0.663 | 0.405 |
| Ref-NPR | 0.164 | 0.122 | 0.273 | 0.126 | 0.289 | 0.471 | 0.669 | 0.596 | 0.339 |
| ReGS | **0.127** | **0.119** | **0.175** | **0.104** | **0.134** | **0.367** | **0.454** | **0.472** | **0.202** |

the original art, whereas our method follows the extract stylized texture in the reference image by design.

# F   More Results

We present more visual comparison results in Figure 14 & 15 to better demonstrate the superiority of our approach. In Figure 16, we show results of multiple style references acting on a single baked scene. In Table 2, we report Ref-LPIPS score of each scene. Our method consistently outperforms baselines.

