# OpenReview forum: "ReGS: Reference-based Controllable Scene Stylization with Gaussian Splatting"
_NeurIPS.cc/2024/Conference — NeurIPS 2024 poster_

### Official Review · Reviewer_JWsE · 2024-07-01

**Soundness:** 2
**Presentation:** 3
**Contribution:** 2
**Rating:** 5
**Confidence:** 3

**Summary:**

This paper presents a method for stylizing 3D Gaussian Splatting (3DGS) using a single reference image. Unlike NeRF, which uses a structured representation, 3DGS is an unstructured discrete representation that tightly binds geometry and appearance to each Gaussian splat. To address this challenge, the paper introduces a texture-guided control mechanism, which differs from the position-guided approach used in the original 3DGS paper. This new mechanism effectively edits the appearance of a pretrained 3DGS to match the detailed texture from the reference image while preserving the original geometric structure.

**Strengths:**

+ The main novelty of this work lies in the Gaussian splitting strategy, which is based on the color gradients of all Gaussians over iterations, rather than the positional gradients used in the original 3GDS approach. I find this approach to be quite neat and well-suited for the task.
+ The ablation study demonstrates the benefits of using this approach, including a reduction in the number of Gaussians needed to model the details of the reference texture.

**Weaknesses:**

+ The novelty of the method seems somewhat limited, as it is largely based on Ref-NPR to enable image-reference-guided stylization.
+ It is unclear how well the method would perform if the geometry is also heavily stylized, rather than just the appearance.
+ The results (specially the video results) presented are quite limited, e focusing primarily on simple synthetic scenes with white backgrounds, and do not demonstrate the method's effectiveness on more complex scenes.

**Questions:**

+ Can this method be applied when the geometry is heavily stylized? Most of the stylization examples seem to focus only on color/appearance.
+ If possible, could the author share the video results for the Fern and Truck scenes in Figure 6?

**Limitations:**

The limitations of the method are adequately addressed, and interesting future directions are proposed.

---

> ### Author Rebuttal · Authors · 2024-08-02
>
> Thank you for your constructive feedback and thoughtful comments! Please note our top-level comment and additional experimental results in the rebuttal PDF. Below we address your questions and concerns.
>
> ---
>
> **wrt to novelty.** The core contribution of this paper is to enable stylizing 3DGS using an aligned reference image, which can benefit many real applications given its real-time rendering speed. **For the first time, we illustrate that the bottleneck of stylization lies in the nature of 3D Gaussians rather than stylization techniques.** Therefore, we made important contributions on this new 3D representation by **(1)** identifying the entangled appearance and geometry of 3DGS is the primary bottleneck, and **(2)** proposing a set of novel techniques to address the entangelement, resulting in **(3)** a complete framework, for the first time, enables real-time stylized view synthesis. These novel designs are crucial for obtaining high-quality renderings. We agree that we follow the style transfer techniques from Ref-NPR but they are perpendicular to our core contributions. We will make our contributions more clear in the revised paper.
>
> ---
>
> **wrt stylizing geometry.**  Following existing works [8, 9, 3, 10, 6], this paper focuses on stylizing the appearance, which remains an open problem for 3DGS. Our work aims to enable precise texture editing based on a content aligned reference. We agree that further stylizing the geometry will be very interesting, but would like to also mention that such task is significantly more challenging given current problem setup, *i.e.* using one reference image, due to the single-view shape ambiguity. Existing methods achieving high-fidelity geometry stylization often require a very different set of techniques such as shape prior$^1$, text guidance$^2$ and/or generative modeling$^{3,4,5}$ that is beyond the scope of this work. Combining our method with these techniques to enable both geometry and appearance stylization for 3DGS is an open and interesting future direction.
>
> ---
>
> **wrt video results on more complex scenes.** Besides synthetic scenes, the demo video also contains two forward-facing scenes (1:20 and 1:33). We will include more complex scenes like Fern and Truck in the updated video. Unfortunately, according to the rebuttal instruction and further confirmed by the AC, we found that it is not possible to upload additional videos to openreview during the rebuttal phase. Instead, in Figure 6 in the rebuttal PDF, we render images from multiple viewpoints of requested scenes as a workaround. Our method can achieve consistent 3D stylization for these more complex scenes.
>
> ---
>
> Reference:
> 1. Bao, Chong, et al. "Sine: Semantic-driven image-based nerf editing with prior-guided editing field."CVPR. 2023.
> 2. Wang, Can, et al. "Nerf-art: Text-driven neural radiance fields stylization." IEEE Transactions on Visualization and Computer Graphics (2023).
> 3. Haque, Ayaan, et al. "Instruct-nerf2nerf: Editing 3d scenes with instructions." ICCV. 2023.
> 4. Chen, Yiwen, et al. "Gaussianeditor: Swift and controllable 3d editing with gaussian splatting." CVPR. 2024.
> 5. Wang, Junjie, et al. "Gaussianeditor: Editing 3d gaussians delicately with text instructions." CVPR. 2024.

---

> > ### Comment · Reviewer_JWsE · 2024-08-11
> >
> > Thank you for your detailed answers.  I appreciate the main novelty proposed by the method and thus, still recommend Borderline Accept. However, I maintain that the scope of the contribution/novelty is quite limited (no geometry stylization, reliance Ref-NPR and the fact that the real time rendering speed is from the choice of using GS itself, rather than contribution from the paper) hence I am not raising my score.

---

> > > ### Author Response · Authors · 2024-08-12
> > > **Thank you!**
> > >
> > > Thank you for your valuable response and assessment. We are glad to see that our main novelty is appreciated.
> > >
> > > Thank you once again for your time and support towards this manuscript.

---

### Official Review · Reviewer_hmWz · 2024-07-06

**Soundness:** 3
**Presentation:** 1
**Contribution:** 2
**Rating:** 6
**Confidence:** 4

**Summary:**

The paper presents an optimization-based approach for style transfer of a (pre-baked) 3D scene represented by a 3D Gaussian splatting (3DGS). In order to fine-tune the given 3D scene with a style reference image of a single view, the authors suggest using a texture-guided controlling algorithm, which modifies the densification algorithm of the original 3DGS by focusing on the color gradients. The training loss is also modified to include depth-based geometry regularization and additional guidance provided by generated pseudo-views based on the 3D projections of the given style reference onto novel view cameras. The experiments are performed upon the existing public weights of 3DGS, where the method is compared with three NeRF-based methods, ARF, SNeRF, and Ref-NPR.

**Strengths:**

1. As demonstrated in the supplementary materials and figures in the manuscript, the method seems to work well with the pre-baked 3DGS weights.
2. Detailed related work section enlightens novice readers to get familiar with the field of style transfer of 3D scenes.
3. Adequate level of implementational details are provided.

**Weaknesses:**

1. Although the topic and the approach presented in the paper seems adequate, the presentation of those can be much better. For example, since the authors have modified the original training algorithm of 3DGS in Section 3.2 of the manuscript, and this seems to be the most significant contribution of this paper, they can use *Algorithmic/Algorithm2E features of LaTeX* or present with *a pseudocode of the densification algorithm* to more clearly present the key differences between theirs and the original 3DGS.
2. The components of the proposed loss functions, such as the TCM, the pseudo-view loss, the depth regularization, and the color-matching loss, which are originally devised to work with NeRF-based scene representation (Ref-NPR). I do not want to argue with the novelty of this adoption, but I believe that the design decision should be more firmly verified. Even though these losses may be generalizable to 3DGS-based representations as the paper implies, this hidden claim should be re-assessed with each component on the compatibility with the new representation (3DGS). In other words, *ablation studies for these loss functions* can be carried out just like [Figure 6 of Ref-NPR paper](https://ref-npr.github.io/assets/2212.02766.pdf) in order to justify the fitness of the proposed loss function with 3DGS representations.
3. I understand that an exhaustive quantitative analysis in this topic can be very difficult to design, but comparing the results with only one table seems not promising enough. For example, detailed tables with each test scene, just like [Table B.1 of Ref-NPR](https://ref-npr.github.io/assets/2212.02766.pdf), can be added with more visualization.
4. The paper could be much better with visualization of *how different style reference images affect a single scene* with the proposed algorithm. For example, Ref-NPR shows results with multiple style inputs acting on a single baked scene.

As a summary, my key concern is the (1) representation of the materials, the (2) justification of the presented/adopted components (the losses, the densification algorithms), the (3) lack of quantitative comparison table of each scene, and the (4) lack of comparison of the results from different style images.

The main contribution of the paper I believe is to report the results from applying the training algorithms of Ref-NPR to 3DGS-based representations with proper algorithmic modification to make it suitable for 3DGS. One requires to compare at least all the cases demonstrated in Ref-NPR in order to justify that this training scheme for style transfer is better suited for 3DGSs than NeRFs. Therefore, unless the mentioned points are addressed, I believe this version of the manuscript is not ready for publication in this venue.

**Questions:**

1. How were the balancing parameters lambda of equation (8) obtained? Are the values of these hyperparameters critical in achieving high quality results, or are the optimal set of values differ across different stylization tasks? If so, providing the recommendation of choosing these hyperparameters will make the work more applicable.
2. Since the approach only densifies (and not prunes, may I guess) the Gaussians, the resulting scene should be much heavier than the original. How much are the number of Gaussians change in the shown experiments? How the quantitative scores (Ref-LPIPS etc.) change as the number of Gaussians increases? Is there any recommendation to modulate the size of the stylized Gaussian splatting?

Please note that these questions are not counted in my overall scoring.

**Limitations:**

Yes, the limitations are adequately addressed in the appendix.

---

> ### Author Rebuttal · Authors · 2024-08-04
>
> Thank you for your valuable comments and constructive feedback! Please note our top-level comment and additional experimental results in the rebuttal PDF. Below we address your questions and concerns.
>
> ---
>
> **wrt presentation of the materials.** Thanks for your constructive suggestion. We have careful revised the manuscript and will keep revising it to improve the presentation. We have included the suggested pseudo code of our densification algorithm using *Algorithm2E features*  in Algorithms 1&2 in the rebuttal PDF. Compared to the original algorithm: ReGS relies on **1.** color gradient instead of positional gradient as the densification guidance, and **2.** uses structured densification, *i.e.,* replacing a parent Gaussian with a structured set of tiny Gaussians instead of two ''medium-sized'' ones in original 3DGS. These designs are crucial for addressing texture underfitting and verified by the experiments (see Sec. 4.2 and Appendix C, and Figure 4 in the rebuttal PDF).
>
> ---
>
> **wrt justification of the adopted loss functions.** We would like to first clarify that **only TCM and color matching loss** are adopted from Ref-NPR. **Depth regularization** and  **pseudo-view loss** are proposed in this work and **have already been studied and ablated in Figure 4 (b)&(c)** in the paper. While TCM and color matching loss are not our main contributions, we agree that re-assess them is necessary and helpful. We report their ablation results in Figure 1 of the rebuttal PDF. As shown, using color matching loss reduces color mismatch and using TCM loss removes artifacts in the occluded areas. These findings are similar to the observations in Ref-NPR. These results suggest that the adopted losses indeed work for our 3DGS-based model. We will include this experiment in the revised paper.
>
> ---
>
> **wrt quantitative comparison table of each scene.** Thanks for your suggestion. Following Table B.1 of Ref-NPR, we report Ref-LPIPS score of each scene in the table below. Our method consistently outperforms baselines. We will include this detailed table in the revised manuscript.
>
>
> | Ref-LPIPS $\downarrow$  | Chair | Ficus | Hotdog | Mic | Flower | Horn | Truck | Playground | Average |
> |:------------------------------------------:|:-----:|:-----:|:------:|:---:|:------:|:----:|:-----:|:-----------:|:-------:|
> | ARF                   |  0.185 | 0.123 | 0.300  | 0.146 | 0.619 | 0.502 | 0.683 | 0.592       | 0.394   |
> | SNeRF             | 0.188 | 0.129 | 0.283  | 0.138 | 0.646 | 0.492 | 0.702 | 0.663       | 0.405   |
> | Ref-NPR      | 0.164 | 0.122 | 0.273  | 0.126 | 0.289 | 0.471 | 0.669 | 0.596       | 0.339 |
> |**ReGS** | **0.127** | **0.119** | **0.175** | **0.104** | **0.134**| **0.367**| **0.454**|**0.472**|**0.202**|
>
> ---
>
> **wrt results for different style images acting on a single scene.** Thanks for your suggestion. We have included results of multiple style references acting on a single baked scene in Figure 2 in the rebuttal PDF. We temporarily omit the references and original content views here due to limited space. We will include more visual results in the revised manuscript.
>
> ---
>
> **wrt balancing parameters of equation (8).** We perform a simple grid search for these hyperparameters on Blender scenes, and empirically found these values generalize well on other scenes and across difference style references. We will include these details in the revised paper.
>
> ---
>
> **wrt number of Gaussians changed during stylization.** Our method maintains the original opacity-based pruning strategy during training, therefore do prunes Gaussians with low opacity. By design, new Gaussians are adaptively created in the texture underfitting areas. The number fully depends on the complexity of the reference texture. Therefore, the resulting scene is **not necessarily heavier** than the original. If the overall reference texture is simpler than the original scene, there will be less Gaussians compared to the original scene. For example, the wood-drums case shown in the Figure 4 in the paper has less Gaussians (0.29M) after stylization than the original scene (0.41M). If the reference texture is more complex, then the resulting scene can be heavier than the original scene.  For example, the mic scene in Figure 1 in the paper has 0.25M more Gaussians after stylization. The quantitative scores will increase as new Gaussians at underfitting areas are created to fill the missing texture, but will remain the static after convergence (*i.e.* no texture underfitting remains). We will include this discussion in the revised paper.

---

> > ### Comment · Reviewer_hmWz · 2024-08-11
> >
> > Thank you for the detailed rebuttal with rich demonstration. Believing that these results and additional discussions will be included in the final manuscript, I do not longer oppose to publication of this work. However, sharing my concerns with the reviewer **JWsE**, the work still seems to be an adoption of Ref-NPR to 3DGS, and thus the scope and the novelty of this work is not significant enough for higher acknowledgement (e.g., awards). I will raise my score accordingly.

---

> > > ### Author Response · Authors · 2024-08-12
> > > **Thank you!**
> > >
> > > Thank you for the valuable comments and elevating the score. We are committed to incorporating the suggested evaluations and discussions in the revised manuscript.
> > >
> > > Thank you once again for your time and assessment.

---

### Official Review · Reviewer_TJDG · 2024-07-09

**Soundness:** 3
**Presentation:** 2
**Contribution:** 3
**Rating:** 6
**Confidence:** 4

**Summary:**

The paper proposes a method to stylize 3D Gaussians using a texture guidance. The method takes a pretrained 3D Gaussian model and one content-aligned reference image as inputs and outputs a stylized 3DGS model which could be rendered at real-time framerate. Several techniques, including structured densification, depth-based geometry regularization and view-consistency constraints are introduced to achieve an effective stylization which performs better than previous state-of-the-art work.

**Strengths:**

1. The paper is generally well-written and easy to follow.
2. The insight on color gradients is interesting and works well. The method seems promising for appearance editing on Gaussians.
3. Both qualitative and quantitative evaluations show noticeable improvement compared to previous work.

**Weaknesses:**

1. The methodology seems largely inspired by Ref-NPR, though adapted to fit the 3D Gaussians. Readers may have to read Ref-NPR first in order to understand the motivation behind the design choices, especially in Section 3.4.
2. The superscript $(x, y)$ in Eq. 5 is not explained.
3. Minor indentation issues on L154, L188, L198, and L224.

**Questions:**

1. As you have mentioned, calculating TCM loss is slow. An ablation may better explain why TCM must be introduced despite its long running time.
2. Is it possible to use multiple views as texture references?

**Limitations:**

Limitations are addressed in the supplemental material.

---

> ### Author Rebuttal · Authors · 2024-08-02
>
> Thank you for your constructive feedback and thoughtful comments! Please note our top-level comment and additional experimental results in the rebuttal PDF. Below we address your questions and concerns.
>
> ---
>
> **wrt inspired by Ref-NPR.**  Thanks for pointing this out. We will include a more detailed description of Ref-NPR in the updated Appendix to help readers understand the motivation behind Sec. 3.4.
>
> ---
>
> **wrt superscript  $(x,y)$ in Eq. 5.** $(x, y)$ denotes the $xy$ coordinates on a feature map. Therefore, Eq. 5 refers to find the location $(x^{\*}, y^{\*})$ on $F_{I_{R}}$ that has closest feature-wise distance to the given location $(x,y)$ on $F_{I_{i}}$.  $(x^{\*}, y^{\*})$  is then used to reconstruct a guidance feature  $F_{G_{i}}$ for TCM (Line 218). We will clarify this in the revised manuscript.
>
> ---
>
> **wrt minor indentation issues.** Thanks for your careful reading. We will fix these issues in the revised paper.
>
> ---
>
> **wrt ablation on TCM loss.** Thanks for your suggestions. As stated in Line 214, TCM loss is used to spread stylized appearance to the occluded areas, so that the entire scene is stylized. We have included an ablation on TCM in Figure 1 (b) in the rebuttal PDF. Without TCM, the model cannot properly stylize the unseen areas and produces artifacts. Applications like appearance editing (Appendix D), that do not require stylizing entire scene, do not need TCM loss. We will include this ablation in the revised paper.
>
> ---
>
> **wrt multi-view texture references.** Yes, our method can be naturally extended to take multi-view texture references. We leave such exploration for future work.

---

> > ### Author Response · Authors · 2024-08-12
> >
> > Thank you again for your valuable comments. We have tried our best to address your questions (see rebuttal PDF and above), and will carefully revise the manuscript by following suggestions from all reviewers.
> >
> > Please kindly let us know if you have any follow-up questions.

---

> > > ### Comment · Reviewer_TJDG · 2024-08-12
> > >
> > > Thank you for your detailed answers. The rebuttal clearly addresses my questions.

---

### Official Review · Reviewer_BehR · 2024-07-10

**Soundness:** 3
**Presentation:** 3
**Contribution:** 3
**Rating:** 7
**Confidence:** 4

**Summary:**

The paper proposed a texture-guided Gaussian densification strategy for exemplar-based 3DGS style transfer with content-aligned reference, while preserving original geometry by depth supervision.
During 3D stylization with a style template reference, the introduced texture-guided Gaussian control strategy can geometrically adaptively densify 3D Gaussians for fine-grained texture optimization.
Relying on the advanced representation of 3DGS, the stylized scene can achieve real-time rendering of novel views.

**Strengths:**

1. The paper proposed a decent design of style transfer for a 3DGS scene while preserving geometry by depth supervision.
The novel texture-guided control of Gaussian densification assists in optimizing texture with high-frequency details.
I believe this strategy worths attention beyond 3DGS appearance stylization.

2. The Stylized Pseudo View Supervision works better than other multi-view consistent stylization baselines, in terms of semantic consistent stylization for uncovered areas by reference view.

3. The elaboration of methodology is technically sound, which is possibly reproduced.

4. The experiments and evaluation are convincing with ablation studies and baseline comparisons. And paper experimented on diverse scenes covering objects, forward-facing scenes, an unbounded scene. But I still have some main concerns mentioned in Weaknesses 2.

**Weaknesses:**

1. The paper mainly concerns fine appearance optimization by densification and depth supervision.
For 3D stylization, geometric stylization and editing could be tried or discussed based on proposed method. For example, stylizatin given an edited view with minor shape changes.

2. The most innovative and inspiring part is the Texture-guided Gaussian Control with texture guidance plus structured desification. However, the experiment part can be further improved:

    2.1. In Appendix C, there an ablation study by comparing original 2 Gaussians and proposed 9 Gaussians densification set. There is no solid and scientific validation for the best selection of the number 9. Please see details in Question 2.

    2.2. There is no ablation study of ablating only texture guidance (i.e. use original position gradients as guidance), or ablating only structured densification (i.e. use original densification scheme). Current Sec 4.2 ablation study of Texture-Guided Control show the joint effect of texture guidance and structured desification, which cannot show the effects come from the joint cooperation or from one dominant strategy. Please see details in Question 3.

3. A minor point and suggestion.
For evaluation comparisons, the paper mainly compare with baselines with Plenoxels representation.
Since ReGS's fast training and rendering capability replies on 3DGS, even ablation studies provide good validataion, I still expect comparisons with baselines with 3DGS, e.g. reproduce 3DGS-version SNeRF.

4. Minor issue in related work section. The paper should stress 2D and 3D stylization involve only image-exemplar based neural style transfer.
Since this work finishes edited-view guided stylization, methods of text-guided dataset editing for optimization such as Instruct-NeRF2NeRF/Instruct-GS2GS is also a suitable related work.
There are also some concurrent work stylizing 3DGS scenes, such as StyleGaussian, StylizedGS, Gaussian Splatting in Style.

**Questions:**

1. For texture guidance, the paper selects color gradient as hints for desification, which is a straightforward constraint hint. Is the selection based on trials among all variables such as scales, colors, rotations, opacity, etc.? If yes, what are differences among different gradient hints?

2. In the ablation study of Structured Densification (in Appendix C), I would suggest to conduct an experiment with different numbers of a dense set of Gaussians for each responsible Gaussian to be splitted, varying from original 2 to proposed 9 or even larger number.
There is not enough experimental statistics to support the densification strategy via replacing by a denser set of 9 Gaussians, rather than smaller 5, or larger 16 Gaussians.
In addition, in Appendix C default setting is based on position-gradient-guided or proposed color-gradient-guided density control?

3. In Texture-guided Gaussian Control, which one between Texture Guidance and Structured Densification is more important? Or only when both jointly work, ReGS can gain better performance than naive densification strategy?

4. I wonder if this Gaussian densification strategy supports original reconstruction and other downstream tasks.

I would like to see more analysis and insights particularly for Questions 1-3 in the discussion phase.

**Limitations:**

The paper discussed limitations and provided some potential solutions.
The paper does not involve potential negative social impact.

---

> ### Author Rebuttal · Authors · 2024-08-02
>
> Thank you for your detailed review and constructive suggestions! Please note our top-level comment and additional experimental results in the rebuttal PDF. Below we address your questions and concerns.
>
> ---
>
> **wrt geometric stylization.** Thanks for your suggestion. We agree that also editting the geometry will be very interesting. We believe our method do able to handle minor shape changes, for example, by relaxing the depth supervision. However, we would like to also mention that precise geometry editting based on a reference image is inherently more challenging due to single-view shape ambiguity.  To achieve high-quality geometry stylization, existing methods often adopt a very different set of techniques such as shape prior$^1$, text guidance$^2$ and/or generative modeling$^{3,4,5}$ to hallucinate missing geometry, which are beyond the scope of this work. Combining our method with these techniques for joint geometry and appearance editing is an open and interesting future direction. We will include more discussions in the revised paper.
>
> ---
>
> **wrt the selection of the number 9 in densification.** Thanks for your suggestion.  We actually conducted a similar experiment when developing this method, and we found the current splitting strategy provides the best overall performance. Here we conducted the suggested ablation on the number of Gaussians and present the results in Figure 3 in the rebuttal PDF. We plot the PSNR value between the style reference and the corresponding stylized view to quantitively show the texture fitting capability using Blender scenes. As shown, when the number is small, the model fails to capture the target texture details. As this number grows, the performance becomes saturated. When the number equals 9, the model can achieve peak performance but also without inducing many excessive Gaussians that might slow down rendering. We will include this experiment in the revised manuscript.
>
> ---
>
>  **wrt the *Default* setting for Appendix C.** The *Default* setting is based on the color-gradient-guided density control (same as *Ours*) and therefore we can fairly ablate structured densification. In the experiment, we showcase the effectiveness of the proposed structured densification by studying how the performance will be affected when we remove such design (*i.e.* by switching to the default strategy) from the full model. We will clarify this in the revised paper.
>
> ---
>
> **wrt ablation on texture guidance.** Thanks for your suggestion. Here we conduct the suggested ablation study on texture guidance, where we construct the baseline by removing texture guidance from the full model (*i.e.*, switching to the default positional-gradient guidance). We report the comparison results in Figure 4 in the rebuttal PDF. As shown, without texture guidance, the model fails to capture tiny texture details in the reference. We will include this experiment in the revised paper.
>
> ---
>
>  **wrt the importance between Texture Guidance and Structured Densification.** From the Appendix C and Figure 4 in the rebuttal PDF, one can see that both strategies are equally important. Removing either of them will reduce stylization quality.
>
> ---
>
> **wrt 3DGS version of baseline SNeRF.** Thank you for the suggestion.  We will reproduce an 3DGS version of SNeRF and use it as an additional baseline the revised paper.
>
> ---
>
> **wrt  related work.** Thanks for your suggestion. We will revise the related work section accordingly.
>
> ---
>
> **wrt other gradient hints.**  Yes, we have tried to use other gradient hints but none of them achieves similar level of details and fidelity as color gradient. This is because other variables (e.g. scales, rotations, and opacity) are not directly/strongly related to final appearance and thus are less sensitive to texture underfitting, especially in the high-frequency areas.
>
> ---
>
> **wrt supports for other tasks.** Our method can potentially benefit scene reconstruction and other downstream tasks, especially when the scene texture is complex. For example, one can simply use our method as an additional training stage to further refine texture details that are missing in the initial reconstruction.  Such exploration might be an interesting direction for future work.
>
> ---
>
> Reference:
> 1. Bao, Chong, et al. "Sine: Semantic-driven image-based nerf editing with prior-guided editing field."CVPR. 2023.
> 2. Wang, Can, et al. "Nerf-art: Text-driven neural radiance fields stylization." IEEE Transactions on Visualization and Computer Graphics (2023).
> 3. Haque, Ayaan, et al. "Instruct-nerf2nerf: Editing 3d scenes with instructions." ICCV. 2023.
> 4. Chen, Yiwen, et al. "Gaussianeditor: Swift and controllable 3d editing with gaussian splatting." CVPR. 2024.
> 5. Wang, Junjie, et al. "Gaussianeditor: Editing 3d gaussians delicately with text instructions." CVPR. 2024.

---

> > ### Comment · Reviewer_BehR · 2024-08-09
> >
> > I thank the authors for the comprehensive rebuttal and additional experiments, which addressed my main concerns in W2.
> >
> > While this work was inspired by Ref-NPR, it presents a successful adaptation to the different and more effective 3D-GS representation. I believe the proposed optimization strategy would arouse significant interest and discussion within the research community. This paper not only serves as one of the earliest works on 3D-GS stylization, but also has the potential to benefit broader research on 3D-GS texture reconstruction/refinement and appearance editing.
> >
> > I would like to elevate my score given these additional experiments and more extensive discussion in the revision.

---

> > > ### Author Response · Authors · 2024-08-09
> > > **Thank you!**
> > >
> > > Thank you for the kind words and elevating the score! We are glad to see our contribution is recognized and hear we have addressed your concerns.

---

### Official Review · Reviewer_3qBC · 2024-07-12

**Soundness:** 3
**Presentation:** 3
**Contribution:** 3
**Rating:** 5
**Confidence:** 4

**Summary:**

The paper introduces ReGS, a new reference-based 3D style transfer method that utilizes 3DGs as the 3D representation. To capture fine-grained details from the reference view, the method employs texture-guided Gaussian control to enhance density in areas where texture is under-represented. Additionally, the approach incorporates depth-based regularization and pseudo-view supervision to ensure consistency while stylizing with the reference image. The quantitative and qualitative results demonstrate that ReGS achieves superior stylization, capturing detailed and high-quality effects more effectively than previous methods.

**Strengths:**

- The paper is well-written and comprehensive, making it easy to follow.
- The experiments are detailed, and the impact of each proposed method is demonstrated step-by-step.
- The stylization results effectively capture fine-grained details from the reference image.
- The proposed appearance-based densification approach is simple yet proves to be effective.
- The choice of 3D GS for reference-based stylization results in faster rendering performance.

**Weaknesses:**

- I do not find the methods presented in the paper to be significantly novel, as they give the impression of being a 3DGS-adapted version of Ref-NPR. While I acknowledge the differences and novelties introduced to effectively adapt reference-based stylization to the 3D-GS setting, I do not see a critical distinction in terms of the 'style transfer technique' itself, once the modifications specific to the 3D-GS settings are set aside. This is primarily because the stylization pipeline (Section 3.4) closely mirrors that of Ref-NPR, without introducing new improvements or modifications.
- The qualitative comparison presented in Figure 6 appears unfair. As I understand, ARF and SNeRF in this experiment are stylized using a stylized reference view, and the discrepancies between these results and the reference view are emphasized. However, the primary objectives of ARF and SNeRF differ from those of Ref-NPR and ReGS, as they are not specifically designed for reference-based stylization. Consequently, there is no inherent need for their stylization results to strictly adhere to the reference view. I believe the authors are aware of this distinction. For a fairer comparison, it would be more appropriate for the authors to include the original 2D style image for ARF and SNeRF and conduct a qualitative assessment based on aesthetic quality. Comparisons of the ability to replicate high-frequency details and correspondence should perhaps be reserved exclusively for comparisons with Ref-NPR.

**Questions:**

Please see the weaknesses above.

**Limitations:**

The authors addressed the limitations in the paper.

---

> ### Author Rebuttal · Authors · 2024-08-02
>
> Thank you for your detailed review and constructive comments! Please note our top-level comment with additional experimental results in the rebuttal PDF. Below we address your questions and concerns.
>
> ---
>
> **wrt novelty on style transfer techniques.** The core contribution of this paper is to enable precise stylization of 3DGS using an aligned reference image, which can benefit many real applications given its real-time rendering speed. **For the first time, we illustrate that the bottleneck of such stylization lies in the nature of 3D Gaussians rather than the well-established style transfer techniques.** Therefore, we made important contributions on improving this new 3D representation by **(1)** identifying the entangled geometry and appearance is the primiary bottleneck for stylization, and proposing **(2)** a set of novel techniques (Sec. 3.2&3.3) to tackle this entanglement, resulting in **(3)** a complete method, for the first time, that enables real-time stylized view sythesis. These novel designs are verfied by extensive experiments (Sec. 4.2&4.3). Style transfer techniques from Ref-NPR are perpendicular to our core contributions. But we agree that these techniques might not be perfect for 3DGS and further improving them is an interesting future direction.
>
> ---
>
> **wrt comparison with ARF and SNeRF on aesthetic quality.** Thanks for your advice. We simply follow Ref-NPR to include ARF and SNeRF as additional baselines. We do aware such distinction and agree that evaluating aesthetic quality using the original style image for them is indeed a fairer comparison. In Figure 5 in the rebuttal PDF, we report the suggested qualitative comparison with SNeRF. The original style images are acquired from Ref-NPR authors. One can see that SNeRF produces results mimicking the abstract style of the original art, whereas our method follows the extract stylized texture in the reference image by design. We will include more visual comparisons in the revised manuscript.

---

> > ### Author Response · Authors · 2024-08-12
> >
> > Thank you again for your valuable comments. We have tried our best to address your questions (see rebuttal PDF and above), and will carefully revise the manuscript by following suggestions from all reviewers.
> >
> > Please kindly let us know if you have any follow-up questions.

---

> > > ### Comment · Reviewer_3qBC · 2024-08-13
> > >
> > > I thank the authors for their rebuttal and the additional results provided. While I find no compelling reason to oppose publication, I acknowledge the paper’s contributions as an effective adaptation of ref-npr to 3DGS representation. Although this represents a novel approach, its overall significance might not be substantial. Thus, I will maintain my current rating.

---

### Author Rebuttal · Authors · 2024-08-02

We would like to thank all reviewers for providing constructive feedback that helped us improved the paper. We are encouraged that the reviewers think
- our approach is decent (BehR), neat (JWsE), and interesting (TJDG)
- designs and insights are effective (3qBC) and works well (BehR, TJDG, hmWz)
- experiments and evaluation are detailed (3qBC), and convincing (BehR)
- the paper is well-written (3qBC) and easy to follow (TJDG)

We have been working diligently on improving the paper on several fronts, addressing the critique.

**Please note we have included figures for the suggested experiments in the rebuttal PDF**.
We address questions and concerns for each reviewer in the comments below.

---

### Decision · Program_Chairs · 2024-09-25

**Decision:**

Accept (poster)

**Comment:**

This paper received 1 accept, 2 weak accepts, and 2 borderline accepts. Reviewers acknowledge that the proposed method is well-designed and effective. Most concerns raised during the review process, such as components of the loss function and the fairness of qualitative comparisons, have been adequately addressed in the authors' rebuttal. However, questions about the significance of the adaptation compared to the Ref-NPR persist. Despite this, reviewers agree that the paper's contributions and solutions merit publication. Therefore, the Area Chair recommends acceptance of this manuscript.